# Educating on Sexuality to Promote Health: Applied Experiences Mainstreaming the Gender and Human Rights Approach

**DOI:** 10.3390/ijerph18052249

**Published:** 2021-02-25

**Authors:** Ana L. Martínez-Abarca, Ana M. Martínez-Pérez

**Affiliations:** 1School of Medicine, Universidad de las Américas, Quito 170124, Ecuador; 2Health Sciences Program, Rey Juan Carlos University, 28032 Madrid, Spain; 3Department of Communication and Sociology, Faculty of Communication, Universidad Rey Juan Carlos, 28943 Madrid, Spain

**Keywords:** comprehensive sexuality education, salutogenic approach, health promotion skills, mainstreaming (gender and human rights)

## Abstract

In the articulation between research and health intervention, we see the need to situate comprehensive sexuality education in the triangle formed by the salutogenic approach, the mainstreaming of gender and human rights, and the development of competencies in health promotion. For this purpose, we present a set of investigations carried out in Ecuador and Spain through a qualitative methodology with the respective health interventions that seek to obtain significant results of the teaching–learning process on sexuality. The field work contemplates situations of violence as a health problem, orienting the intervention in health empowerment toward pleasure. The health education experiences implemented allow us to conclude that comprehensive sexuality education reinforces the competencies of health personnel to attend to diversity. These findings, and the results expected in ongoing research, allow us to form a competency framework aimed at specifically improving medical education so that medical professionals can generate health processes with a cross-cutting approach to gender and human rights.

## 1. Introduction

Talking about sexuality is complex for most of the population; nevertheless, a comprehensive education on sexuality is essential to enjoy good health [1]. From the biological approach, the social, cultural, and historical implications position knowledge of the body in a distant and unknown place that hides health problems, including discrimination and violence.

The historical differences between sex and gender addressed by feminist theory [2] show the importance of the study of sexuality from a new point of view that includes sexual identity as a performative expression and understand it as a mechanism of power abuse that is presented in these applied experiences through violence. The approach of pleasure is the salutogenic strategy that allows us to apply theories about sexuality in the improvement of health and well-being, as we show in this article [3].

Therefore, the approach to comprehensive sexuality education, from now on CSE, as a public policy has been established in regions such as Latin America, given the need to reformulate knowledge about sexuality, where the focus on gender, health, and pleasure are fundamental aspects that can address fundamental issues such as child pregnancy, sexual abuse, and sexual health [4]. Comprehensive sexuality education includes training on formal and non-formal contexts based on scientific evidence appropriate to each stage and age, which is comprehensive and transformative and then generates health, but it also has a gender and human rights perspective [5]. 

Currently, worldwide, medical schools debate the importance of gender mainstreaming in medical education and training as a tool to guarantee human rights and equality in medical care [6]. The incorporation of gender mainstreaming in the education of health professionals has focused on four main areas: the relationship of gender with cardiovascular health, gender and violence, gender and mental health, and gender and sexuality [7,8].

A study carried out in Colombia reflects the need to incorporate the study of sexuality from a gender perspective in medical education as a tool to guarantee human rights, especially sexual and reproductive rights. This study indicates that if the approach to integral sexuality is not part of the education and training process and is included as a requirement in the professional profile, important changes in care will not be achieved [9,10]. In both the European and Latin American contexts, there are research and applied health intervention groups that seek to improve comprehensive sexuality education. In some cases, emphasis is placed on addressing different forms of sexual violence [11]; in others, discourses on desire and pleasure are analyzed [10]. UNESCO’s institutional campaign “Comprehensive sexuality education: a basis for life and love” aims toward addressing relationships between mothers/fathers and sons/daughters and shared learning about sexuality [5].

The approaches in the studies in Latin America and Europe oriented to comprehensive sexuality education from a gender perspective apply the qualitative methodology of data collection and analysis, including the perceptions of professionals, patients, and citizens. A retrospective study developed in Ecuador between 1998 and 2017 on the situation of sexuality education reflects the importance of deepening the analysis of individual management experiences in populations that seek to guarantee access to rights and the eradication of gender inequalities in the exercise of sexuality. In Ecuador, sexuality education policies have a strong political and not very technical component [11].

In 2021, in Ecuador, United Nations organizations such as UNESCO or UNFPA, in alliance with the Ministry of Education, published a strategic plan for the incorporation of comprehensive sexuality education based on the fulfillment of sustainable development goals and a comprehensive educational agenda until 2030 [12], where the importance of gender and human rights mainstreaming in educational transformation is evidenced.

From the gender point of view, sexuality transcends the biological binary of sex, placing the body in a context of social relations, crossed by culture as a reflection in education public policy. This approach is aimed at working from education on the prevention and eradication of gender-based violence, including the prevention of teenage pregnancy and sexual violence in the classroom as well as the empowerment and mainstreaming of sexual and reproductive rights, which focuses on the needs of each age group either in basic education or high school [13].

Although the models at a descriptive level are not identical, in this article, they are taken as synonyms, as they both respond to the characteristics described above, with a positive positioning toward sexuality that accepts desire and pleasure as essential parts of sexuality. In particular, the importance of mainstreaming gender and human rights in medical education and in health professionals’ education is confirmed by some working group publications that support with scientific evidence the necessary reform of curricula [8,9,14]. Concerning the content of adequate comprehensive sexuality education, the relevance would be in improving the health of the population, including the good overall health of health workers, and considering that sexuality is formed by “individual practices, values, and cultural norms” [15]. However, our reflection goes further, given that the advances of previous research lead us to consider this combined approach of gender and human rights as a requirement for those who work with people in the fulfillment of constitutionally recognized fundamental rights—that is, health, education, and justice [16].

Several research questions guide the work presented here: Are there any lessons that can be useful (even enforceable) to improve clinical health care? Are the rights of the population using the health systems being violated when these training deficiencies occur? What methodology and learning objectives would we have to incorporate into the medical curriculum to ensure adequate health care with a gender and human rights approach? In this sense, our objective with the research we present is to apply the comprehensive sexuality education model to develop health promotion competencies [17] with a salutogenic approach [18] based on gender and human rights [19]. This conceptual framework [7] allows us to develop health empowerment in an applied way both in those who are trained in the classrooms where we teach in Ecuador and Spain (Faculty of Health Sciences, UDLA; Faculty of Communication Sciences, URJC) and in the intervention workshops that we carry out in different institutions.

## 2. Materials and Methods

The work has been developed in two phases. Each of them has the methodology adapted to the research objectives. First, we wanted to know the relevance of a salutogenic CSE model that allows an analysis of violence in its relationship with sexuality and that has an orientation toward pleasure as a health asset. Secondly, we wanted to show the feasibility of the theoretical model of CSE with workshops focused on group dynamics aimed at raising awareness of our sexuality as an essential part of health promotion. In the research phase, we mainly use an ethnographic method based on in-depth interviews (narratives) and participant observation registered in a fieldwork diary. In the intervention phase, we carried out workshops based on group dynamics.

The methodology used in this study, both in the basic research phases and in the intervention applied to health education and promotion, is eminently qualitative and based on the ethnographic method. However, we have taken the updated quantitative data from each country as secondary sources of research for the analysis. In total, we have carried out 20 in-depth interviews for the composition of narratives and more than one hundred days of participant observation recorded through field diaries. The appropriateness of these techniques to the objectives is due to the interest in analyzing in depth the situations lived around sexuality to relate them with the most relevant components in comprehensive sexuality education [1]. It is also fundamental to consider that the two skills on which these techniques are based, active listening and careful observation, are basic to the training of health personnel in which we are involved as higher education lecturers. The limitations of the questionnaire or a closed interview lie in the “inability to capture the symbolic structures and pre-conscious complexities” [20]. Indeed, we use quantitative data as secondary sources that complement the validity for the significance of the primary sources generated and triangulated (intrinsically and extrinsically to the technique itself) in the research process. The selection of informants for the interviews that make up the narratives is done through a snowball process, taking into account the importance of the first-person perspective [21].

Fieldwork has been carried out in Ecuador and Spain in different phases between 2018 and 2020 (as summarized in Table 1) focusing on issues that allowed us to understand the implications of violence against women and girls and its relationship with the skills that health professionals should have to care for them adequately. These thematic investigations lead us to test the initial hypothesis that an improvement in comprehensive sexuality education would improve the quality of health personnel training and, therefore, the health of the population served by the health systems. The qualitative research methodology is justified by the search for results that explain the meanings of sexual health behavior: decision making about one’s own sexuality, problems derived from violence, and competencies for health care appropriate to gender diversity. Talking about violence in its relationship with sexuality is not easy, so we have opted for a qualitative research methodology that allows us to go deeper into the processes experienced and their meanings. The use of specific software for qualitative data analysis, such as MaxQDA, allows us to reach discursive positions through the analysis of informants’ discourses, which we then use in an inter-method triangulation in the groups carried out in the intervention (workshops).

In terms of health intervention, during the three years of fieldwork, 2018, 2019, and 2020, workshops have been held with different population groups in Quito, Ecuador, and Madrid, Spain, as shown in Table 2. Throughout 2018, medical students in the subjects of Medical Anthropology and Sociology and Bioethics at the UDLA, women, and men aged 18–22, received complimentary training in comprehensive sexuality education intending to empower them in health and make them better clinicians in serving the population. During 2019, women between the ages of 25 and 60 attended workshops voluntarily in different Equality Spaces in Madrid City Council and associations linked to different social movements in civil society (activism in feminism, ecology, popular education, community health). Finally, in 2020, workshops were held with men and women aged 30–45, as part of an intercultural mediation program run by the non-governmental organization Doctors of the World, as part of a project to prevent female genital mutilation in girls and adolescents of African origin living in Spain.

## 3. Results

We can find three main findings of this research and the interventions derived from it. First, it is pertinent to approach CSE from the analysis of violence and with the salutogenic orientation of pleasure both for the general population and in the training context of health professionals. The association between violence and sexuality is well established in our cultures; hence, in the teaching–learning processes, it is necessary to place emphasis on differentiating, for example, pornography and eroticism. From the close relationship between violence and sexuality, we construct preconceived ideas that only through an analysis of culture can we deconstruct toward a new evidence-based conceptual framework of CSE [5].

Second, cultural barriers to understanding sexuality with a salutogenic and gender–human rights approach are situated in that the biomedical system is disease-oriented and therefore explanatory frameworks of sexuality go through the risk of contracting sexually transmitted diseases. For the most part, moralistic, risk or biocentric approaches [11] focus on pathology, and only CSE includes pleasure as one of the relevant aspects that interacts with violence but also with sexual rights. This integrative view of sexuality allows, as we hope to confirm in further research, better attention to diversity by health professionals.

Thirdly and finally, sexuality seen as a human behavior or a total social fact, as Durkheim would say, is not integrated in the curricula of the educational systems of Ecuador and Spain. We have come to verify the validity of a didactic method that incorporates an analysis of sexuality based on scientific evidence but also on the awareness of one’s own reality through the use of the narratives that each student constructs in the workshops.

As the manuals consulted refer [5], we cannot analyze sexuality without including violence with updated data from the two countries as secondary sources [13,15]. Even more, focusing on violence makes it possible to deconstruct some concepts that are well established in the culture and that the lack of comprehensive sexuality education does not help to unravel. Two of the essential components of CSE are violence and pleasure [1]; hence, we chose to apply results obtained in previous research [22,23] with the aim that our intervention in comprehensive sexuality education would function as a health asset in itself. We know from interventions carried out in other fields of action that collective modes of generating situated knowledge, for example around the feeling of guilt [24], generate salutogenic processes because the group “serves to know that we are not alone and we can help each other” (Woman, 56 years old, participant in the Anthropology of Health and Life Workshop, Madrid).

In the case of the first year of workshops with medical students, the dramatically high figures of unwanted pregnancy in girls and adolescents led us to introduce specific training in the understanding of the taboo of incest, from an anthropological perspective and then to provide clinical care tools within a framework of respect for human rights and with a gender focus. In fact, according to the Guide to Care for Adolescents [16], suicide is the leading cause of death among adolescent women in Ecuador. Therefore, with the data provided more recently in the report Stolen Lives II [14], we designed a complete training program for future medical professionals so that the care of this age group, given the aforementioned morbidity and mortality, would have the best standards of “warmth and quality”, as the advertising slogan of the Ministry of Public Health said at the time. After explaining the taboo of incest as the universal anthropological taboo that forbids sexual relations in kinship by consanguinity, we started a debate about the fact that it is a taboo, and therefore, there are cultural practices aimed at hiding these behaviors. In the central Andes of Ecuador, the symbolic thinking of the Kichwa population leads to the consideration that girls after menarche live with the “risk” that “if they pass under the rainbow” they can become pregnant (Interview 6, Fieldwork Diary, February 2018).

If we consider that violence is “any avoidable suffering in human beings”, we can think of systemic violence understood as any institutional/cultural procedure that produces an adverse effect and that cannot be attributed to any person but all or a majority [25]. In this way, when medical students begin their rotating internship in gynecology or during the rural health social service as doctors who have already graduated, they may frequently encounter the scene of a male relative accompanying a girl to the primary care center to have her explored because she presents discomfort of gynecological origin. Health professionals in Ecuador have the obligation under the protocol to report sexual abuse or rape of children and adolescents and to care for these people without violating the right to privacy; however, too often, these accompanying relatives do not want to leave the clinic for fear of being reported (Field Journal, April 2018). This situation of conflict is not always resolved in a way that complies with the law, according to the interviews conducted by medical professionals (Interview 12, Interview 15).

Female genital mutilation is only one of the manifestations of violence from the socio-cultural system against women and girls that we addressed in the workshops. We also look at early unions or forced marriages, virginity testing, trafficking for sexual exploitation, unsafe abortion, and other forms of violence, sometimes protected, condoned, or encouraged by the same health system, such as the genital mutilation practiced in hospitals in countries such as Egypt or the obstetric–gynecological violence reported in Ecuador [26] and Spain [27]. In the case of some sexual health problems such as anorgasmia, the same participants in the workshops and before the informants interviewed are the first reason for consultation in sexology when working on health empowerment with a salutogenic orientation [28]. They conclude that “it may be related to a lack of knowledge about one’s own and another’s pleasure” (Fieldwork diary, October 2020) but also to the idea that, traditionally, “almost everything related to reproduction was considered suspicious; especially pleasure” [29]. In the same way, good comprehensive sexuality education would be acting as a preventive measure in the growing processes of violence against women that the consumption of pornography, which is increasingly early and massive, makes more complex. This was the purpose of the Channel 4 series, *Mums make porn*, produced by Firecracker, which serves as an educational tool in SSC workshops. This audiovisual production is oriented toward health [18] because it does not align sexuality with violence, as many of the films these five mothers discover their teenage children are watching. In fact, the experience is useful as a teaching tool for CSE because it shows that other pornography is possible if it is associated with eroticism and considers mutual respect and consent.

In the case of the workshops aimed at preventing and caring for women who have suffered some form of genital mutilation, we took the option of developing a salutogenic CSE that understood pleasure as a health asset. The key ideas in the UNESCO manual as learning objectives for the 15–18+ age group are “consent as pleasurable sexual behavior” and that “engaging in sexual behavior should be pleasurable” (p. 59 and 75) [5]. The conceptual framework that describes CSE as an educational and intervention tool has given us the optimal results that we expected according to the research objectives, as it contemplates the need for intimacy in the processes associated with sexuality, the fact that sexuality is a social construction, that sexuality is linked to power and gender, and finally, that it is a reality that we experience throughout our lives [5]. As the manual itself says, “Sexuality can be understood as a central dimension of the human being that includes biological, social, psychological, spiritual, religious, political, legal, historical, ethical, cultural dimensions that evolve throughout a life” (p. 17). In the case of medical education and the training of health professionals in general, “their curriculum is a lived experience” [30], and if they are trained in hierarchies of institutionalized violence, they can be expected to reproduce pathogenic models, as much as a salutogenic orientation, will facilitate health promotion.

Moreover, in the case of workshops on prevention/care for people at risk/situation of genital mutilation, pleasure orientation is almost more than a salutogenic strategy; it is a recognition of survival and a resource of widespread resistance [28]. We start from [31] Alston’s definition of pleasure as “an experience which, from the moment one has it, one would rather have than not have”. This open conception of pleasure is easy to understand and allows in health empowerment to guide the reflection and rehearse a decision making toward the promotion of one’s own health. In health intervention work with female genital mutilation survivors, decoupling patriarchy from sexuality and deconstructing sexual desire was a challenge. When asked what knowledge we took away from the workshop, “I didn’t know that if my husband wants to have sex with me and I don’t, that’s rape” (Interview 7), “when I feel like it, I feel an itch down there, and that’s not bad” (Interview 9), “now I know that cutting is not done so that we don’t go with other men, which is what I’ve always been told” (Interview 8). These statements were made in the interviews conducted after having learned basic aspects about the anatomical description of the clitoris and the physiological process of pleasure. The conclusions reached by the women reinforce the self-management of their own health and at the same time allow them to expand the possibilities of carrying out comprehensive sexuality education with their sons and daughters (Workshop on prevention of gender violence in people at risk/situation of genital mutilation). Behind the masks of the half-covered faces during the pandemic (Autumn 2020), the women participating in the workshop smile when we tell them the clinical case of a woman with a spinal cord injury who can have orgasms, but also when they discover that the clitoris is larger than the glans, and pleasure is held in the brain (Fieldwork Diary, October 2020). The educational unit dedicated to pleasure begins with the reading of the Universal Declaration of Sexual Rights, which, at the 13th World Congress of Sexology, held in 1997, described eleven rights including the right to sexual pleasure.

Among the medical students who have participated in this research, the benefits of pleasure in general, and sexual pleasure, are listed as active in health in that: “it allows improvement in overall health from the moment it acts as a protector of the cardiovascular function, improves tolerance to stress, the immune response, the response to pain, etc.” (Fieldwork diary, January 2018). We should bear in mind that this research has been carried out in two countries with a Judeo-Christian cultural tradition, which has implications because as one interviewee told us: “The main critical knot is the interference of religious morality in the approach to promoting sexual and reproductive rights among young people and adolescents” (Interview, 17).

From the fieldwork of the research period, a series of conceptions about sexuality emerged that in the emic discourse could be incontestable, but from the intervention of the second phase of this project and in an etic reading, they are presented as a priori to be deconstructed. The key concepts and learning objectives of CSE have allowed us during the intervention to dismantle some statements that our informants and later participants in the workshops had. The process of health empowerment was constructed in terms of responding with recent scientific evidence to the myths that come from symbolic thinking and are present in the emic discourse around sexuality. With students, the use of statements that contradict those obtained in the fieldwork has proved to be successful. In this methodological triangulation, these phrases are used in a group teaching proposal of true or false, which should end with a reasoned response according to the scientific evidence consulted. Some of the statements from the fieldwork that we used in the workshops have been validated as a didactic tool to generate debate because it is not always easy to disassociate the patriarchal vision of sexuality: “Virginity does not exist: the hymen is sometimes broken while riding a bicycle”; “Size does not matter: the nerve endings that allow us to feel pleasure are not in the cervix because if it were, it could not function as a birth canal” (Fieldwork diary, February 2019).

## 4. Discussion

Comprehensive sexuality education as part of the strategies for gender mainstreaming and human rights approach in education and medical training are identified as key elements in improving the quality of health care and as a tool for empowering health professionals and society. However, this process is not enough if we do not include comprehensive sexuality educational policies that promote the exercise of sexuality from the claim of pleasure and the prevention of all forms of violence in other stages of education programs [32].

The process of training new health professionals is one of the main tools for improving the quality of care, especially from the perspective of guaranteeing human rights and health promotion. A series of competencies have been defined and validated in the fieldwork to shape a competency framework aimed at improving the training of health professionals, especially medical professionals, on whose approach we are focusing a doctoral thesis that will be presented at the URJC, where gender mainstreaming and human rights in medical education proposes rethinking medical epistemologies toward the medical humanities and social sciences; hence, this analysis is an interdisciplinary work from medicine, philosophy, anthropology, and sociology. There is a sense of coherence in gender and human rights mainstreaming in medical education that introduces the manageability, significance, and applicability of the salutogenic approach [3], starting from the awareness of gender inequalities, the prevention of all forms of violence, and the promotion of pleasure in the exercise of sexuality as instruments for the guarantee of human rights [33].

The Essential Requirements for Medical Education, established by the Central Committee of the Institute of Medical Education [34], specifies the need for “professional values, attitudes and behaviors, and ethics” to be placed on an equal footing with scientific foundations or clinical skills. This equation coincides with the statements of professionals and students in their final years of medical school when they point out that all issues related to human rights are necessary from an integral health care perspective.

In the studies carried out in the Netherlands [35] and Colombia [9], the proposed strategies point toward the construction of an academic curriculum that includes transversal elements of gender and human rights whose objective is to draw effective attention to health problems related to sexuality, pleasure, and violence through empowerment and promotion.

The health promotion axis must be built from the new epistemological and methodological strategies that allow us to look at the social determinants of health from a gender and human rights perspective. There are methodological strategies that allow the gender approach to be mainstreamed into medical education through information, communication, and care, and which can be implemented at any time in the curriculum [5]. The information allows the clarification of basic concepts of gender studies in health—for example, the difference between sex and gender, the impact of gender on health, and its relationship with other determinants that allow for the understanding of the dimension of health as a biopsychosocial universe and from the salutogenic [3], health-promoting, and violence-preventive perspective. Communication strengthens the components of the doctor–patient relationship with an emphasis on empathy and shared decision-making. Finally, attention is focused on planning preventive, therapeutic, and support strategies for patients [7].

## 5. Conclusions

Talking about gender implies looking at the way society relates to each other through the analysis of the dynamics of power between men and women and how relationships are built and negotiated and also how this mainly generates situations of inequality and inequity. Although gender studies were developed by feminist academics, from a feminist perspective, we can analyze other aspects of social life, such as human rights, diversities, the environment, the economy, development, and of course education and health. Gender and human rights mainstreaming in health implies redesigning current academic and institutional structures and systems in which health care professionals are trained to transform the way that they address sexual and reproductive health issues from a salutogenic view, which will affect social dynamics and the way people perceive and exercise their sexuality. Hence, this research develops a roadmap toward impact assessment of competencies developed in medical education toward gender and sexual and reproductive rights. Including gender mainstreaming as a training tool for health professionals as an important step forward to comprehensive, ethical, and human rights-oriented health care.

Finally, in the application, gender and human rights mainstreaming approach as a new paradigm in medical education supports a doctoral research in Ecuador where there are no similar studies; however, in Europe, Netherland universities already have implemented these strategies with encouraging results. Using the approach in Latin American countries implies recognizing the current paternalistic and patriarchal medical model as obsolete, to rethink the new model with curricular plans responding to the population needs and to the new technological and social challenges that guarantee the elimination of gaps in access and quality of health care. Our epistemology or final aim of this research and the doctoral thesis in progress respond to the working hypothesis that with training in medical education with a gender and human rights focus, health professionals will be able to care for the population with adequate consideration of gender, socio-economic, ethnicity, and age diversity.

## Figures and Tables

**Table 1 ijerph-18-02249-t001:** Research fieldwork.

Research	Incest Taboo (2018)	Violence Occurring or Detected in the Educational System (2019)	Early Childhood, Teenage Pregnancy, Suicide, and Sexual Violence (2020)	Female Genital Mutilation (2020)
Interviews/Narratives	4	6	5	5
Persons interviewed	Medical professionals/students	Teachers/teens in the educational unit	Social and health professionals	Women FGM survivors
Place	Quito, Ecuador			Madrid, Spain

**Table 2 ijerph-18-02249-t002:** Applied research or intervention.

Intervention Phase	I: The Year 2018	II: The Year 2019	III: The Year 2020
Participating group	Workshops on Bioethics and Medical Anthropology (Medical students)	Anthropology of health and life workshop	Workshop on the prevention of gender-based violence in people at risk/gender-based mutilation
Gender	Men and women	Women	Women and men
Age	18–22	25–60	30–50
Venue	University of the Americas, Quito, Ecuador	Spaces for equality, Madrid City Council and Associations	NGO, Doctors of the World-Madrid
Key concepts	Prevention of systemic violence, Health Promotion as a Pleasure		

## Data Availability

We keep a copy of the data referred in digital files of the audio recordings and transcriptions. Together with the field diaries used in the process, these data are kept in our personal archives with due guarantees of protection and confidentiality.

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
