# Peer review of "Educating on Sexuality to Promote Health: Applied Experiences Mainstreaming the Gender and Human Rights Approach"

_ijerph, 2021, doi:10.3390/ijerph18052249_

Round 1

Reviewer 1 Report

  • the conceptual framework with violence and pleasure along a continuum for the study of sexuality is well-constructed
  • the paper does an excellent job of detailing the approach for the workshops and offers insights that are compelling, however, grounding the empirical with some theoretical feminist examinations about sexuality, knowledge formation around sexuality and how violence in many forms can trouble frameworks and alter lives
  • it was so refreshing to hear from the participants of the workshops and it would be beneficial if their voices are more extensively incorporated 
  • getting more detailed information on the research design will also be beneficial, such as the motivations of the study, how participants are recruited etc.

Reviewer 2 Report

I want to thank you for the opportunity to review this manuscript. The time spent creating and sending it is greatly appreciated. In my humble opinion, there are currently some issues with the manuscript that need to be addressed and repaired. Here are my recommendations:

The introduction presents a great scarcity and absence of scientific investigations that make known the current state of the subject under investigation and justify the performance of said work. In addition, to add the hypotheses of the same.

The methodology is not explicit enough, to be able to follow and know the procedure carried out during the work. It should indicate whether the use of any statistical software was used for the analysis of the transcripts.

In the discussion section, you should make a comparison of the results found with those of other authors previously mentioned in the introduction. Currently only three investigations have been cited and these have not been previously cited in the introduction. For this reason, I suggest that the discussion be reviewed and related to the theoretical framework and hypothesis of the study. It would be more informative for readers to connect the results with some previous studies and discuss what are some implications for research and practice. What's new in your study?

In the conclusion section, citations should not be included. Furthermore, it could be expanded to include and / or give examples of how this information can be useful for the population and the health field. Authors must make explicit reference to the practical application of the results obtained (this is one of the strengths of the manuscript), paying special attention to the possibilities offered by the data for the design of future interventions.

Thanks for your contribution.

Reviewer 3 Report

It is urgent that health professionals assume a more comprehensive approach to sexuality that takes into account a gender perspective and a human rights-based approach.

In this sense, the proposal presented is interesting. In addition, it is important that the authors use qualitative methodologies when carrying out their research work.

However, the work presented has important limitations that are difficult to overcome.

In the first place, the authors start from a model of integral education in sexuality; however, they put all their focus on violence: incest, sexual violence (sic), unwanted pregnancies, female genital mutilation. Discourse and practice must be coherent and here they do not seem to be so.

On the other hand, the methodology is very poorly explained and it is not known what aspects have been extracted from the interviews and the working groups.

Finally, in the results there are no significant contributions to the construction of a more significant framework of human sexuality.

Round 2

Reviewer 1 Report

Thank you for making the suggested revisions and for offering this contribution.

Author Response

Thank you for your comments.

Reviewer 2 Report

Dear authors, although you have made various changes to improve your manuscript. This continues to present some problems that must be taken into account and repaired:
Although the introduction has been expanded, the number of studies cited to justify and substantiate the objective research topic remain scarce and insufficient.
And on the other hand, in the conclusion section, citations should not be included.
Thanks for your contribution.

Author Response

Revision have been made on the manuscript, please download which for check. Sincerely thank you for your comments.

Reviewer 3 Report

The authors have done an important work to redo the work presented and adapt it to the requirements made by this reviewer.

Author Response

Thank you for your comments.